# Synthesized PANI/CeO_2_ Nanocomposite Films for Enhanced Anti-Corrosion Performance

**DOI:** 10.3390/nano14060526

**Published:** 2024-03-15

**Authors:** Ahmad M. Alsaad, Mohannad Al-Hmoud, Taha M. Rababah, Mohammad W. Marashdeh, Mamduh J. Aljaafreh, Sharif Abu Alrub, Ayed Binzowaimil, Ahmad Telfah

**Affiliations:** 1Department of Physics, Jordan University of Science and Technology, P.O. Box 3030, Irbid 22110, Jordan; 2Department of Physics, College of Science, Imam Mohammad Ibn Saud Islamic University (IMSIU), P.O. Box 90950, Riyadh 11623, Saudi Arabia; mmalhmoud@imamu.edu.sa (M.A.-H.); mwmarashdeh@imamu.edu.sa (M.W.M.); maljaafreh@imamu.edu.sa (M.J.A.); snabualrub@imamu.sa (S.A.A.); ambinzowaimil@imamu.edu.sa (A.B.); 3Department of Nutrition and Food Technology, Faculty of Agriculture, Jordan University of Science and Technology, P.O. Box 3030, Irbid 22110, Jordan; trababah@just.edu.jo; 4Department of Physics, University of Nebraska at Omaha, Omaha, NE 68182, USA; a.telfah@ju.edu.jo; 5Nanotechnology Center, The University of Jordan, Amman 11942, Jordan

**Keywords:** corrosion protection, protonated polyaniline (PANI), cerium dioxide (CeO_2_), optical bandgap energy, electrical conductivity

## Abstract

This study introduces a novel nanocomposite coating composed of PANI/CeO_2_ nanocomposite films, aimed at addressing corrosion protection needs. Analysis through FTIR spectra and XRD patterns confirms the successful formation of the nanocomposite films. Notably, the PANI/CeO_2_ nanocomposite films exhibit a hydrophilic nature. The bandgap energy of the PANI composite film is measured to be 3.74 eV, while the introduction of CeO_2_ NPs into the PANI matrix reduces the bandgap energy to 3.67 eV. Furthermore, the electrical conductivity of the PANI composite film is observed to be 0.40 S·cm^−1^, with the incorporation of CeO_2_ NPs leading to an increase in electrical conductivity to 1.07 S·cm^−1^. To evaluate its efficacy, electrochemical measurements were conducted to assess the corrosion protection performance. Results indicate a high protection efficiency of 92.25% for the PANI/CeO_2_ nanocomposite film.

## 1. Introduction

Corrosion of metals has been one of the essential problems that faces scientific and industrial societies. Corrosion exerts significant economic and environmental consequences across several divisions of worldwide infrastructure and metal-based properties. Beyond risking public security and prompting considerable destruction, deterioration dislocates operations, dictating wide-ranging asset renovation and replacement. Universally, erosion perpetrates over USD 2.5 trillion in compensations annually, corresponding to 3.4% of the global gross domestic product (GDP) [1,2]. There is a lot of consideration from these societies to improve long-life corrosion protection coating [3,4]. Plans for corrosion inhibition comprehend miscellaneous tactics, including proper design, engagement of corrosion-inhabitant alloys, application of anticorrosion coatings, use of corrosion inhibitors, adoption of cathodic fortification techniques, anodic passivation methods, salt removal procedures, and regular cleaning practices [5,6]. Amongst these methods, anticorrosion coatings are accounted as an efficient, versatile, cost-effective, and up-front solution. Remarkably, industries in China assign 66.5% of their corrosion prevention budget towards anticorrosion coatings, underscoring their prominence [7].

Anticorrosion coatings offer twofold approaches for fortification of materials. First, they offer passive protection by acting as physical barriers in contrast to corrosive agents, illustrated by polymeric dyes and Al_2_O_3_ coatings [8,9]. The barrier characteristics and ionic response of these coatings analytically impact their corrosion protection efficacy [10,11,12]. However, passive protection’s operational effectiveness depends on the coating’s physical integrity, which does not warranty optimum substrate protection. Corrosive agents can infiltrate intact coatings through pores, free volumes, and channels, attaining the substrate surface. Secondly, anticorrosion coatings provide an energetic shield by integrating corrosion inhibitors or compensating materials into their preparations to hinder or halt corrosion. Examples include zinc-rich coatings and smart coatings, capable of safeguarding the substrate metal post-damage [13]

Polyaniline (PANI) is one of the most promising conductive polymers that is used in the fabrication of corrosion protective coating due to its good redox recyclability, variable electrical conductivity, and good environmental stability [14]. PANI nanocomposites are used as an exhilarating technique to enhance the corrosion protection performance significantly. This could be attributed to the fact that PANI composite coating triggers the redox action in an extensive range of pH values and increases corrosion protection performance [15,16,17].

Several works have elaborated on techniques to enhance the corrosion protection of PANI coating by introducing ceria nanoparticles (CeO_2_ NPs) [18]. M. Hosseini and K. Aboutalebi [19] reported that incorporating epoxy coating with CeO_2_@PANI@MBT significantly enhances the corrosion protection performance of mild steel substrates. In addition, M. Montemor et al. [20] and L. Calado et al. [21] reported that modifying silane film and siloxane film by CeO_2_NPs enhances the corrosion protection and film stability. Y. Lei et al. [22] concluded that PANI/CeO_2_ nanocomposite coating could be a potential candidate to improve the protection performance of the epoxy coating on carbon steel.

In this research, we introduce an innovative nanocomposite coating comprising PANI/CeO_2_ nanocomposite films tailored for corrosion protection applications. We delve into a comprehensive analysis of the chemical, structural, morphological, and surface wettability properties of these PANI/CeO_2_ nanocomposite films. Furthermore, we explore the impact of incorporating CeO_2_ into the PANI matrix on the thermal stability of the composite. Detailed investigations into the optical and electrical properties, including absorption coefficient, bandgap energy, and electrical conductivity, provide valuable insights. Additionally, we conduct a thorough examination of the corrosion protection performance of PANI/CeO_2_ nanocomposite films across varying temperatures, shedding light on their efficacy in real-world conditions.

## 2. Corrosion Protection

### 2.1. Corrosion Mechanism

Corrosion signifies the procedure whereby metals undertake chemical or electrochemical reactions with their surroundings, leading to their conversion into more chemically stable oxides. This phenomenon contains the electrochemical oxidation of metals in reaction with various oxidants, such as oxygen, hydrogen, or hydroxide. Corrosion formation, demonstrated by the creation of iron oxides, assists as a conspicuous sketch of electrochemical corrosion, classically categorized by the occurrence of a distinctive orange hue. The detrimental effects of corrosion spread to the dilapidation of the material and structural properties, covering mechanical strength, appearance, and resistance to the pervasion of liquids and gases [23]. The corrosion susceptibility of structural alloys is often delicate with contact to atmospheric moisture, although this process can be expressively influenced by the existence of specific substances. Corrosion manifestations may range from localized concentrations, resulting in colliery or fissure formation, to more diffuse, unchanging degradation across the surface. Given that corrosion is primarily a diffusion-driven phenomenon, it mainly takes place on exposed surfaces. Accordingly, techniques intended at weakening the activity of uncovered surfaces, such as passivation and chromate conversion, serve to enlarge a material’s resistance to corrosion [24].

From a chemical standpoint, corrosion can be explicated as an electrochemical process wherein oxidation occurs at a distinct location on the metal surface, functioning as the anode. Electrons injected at this anodic site traverse through the metal to an alternative location on the object, where reduction of oxygen occurs, expedited by the manifestation of hydrogen ions (H^+^). These hydrogen ions may originate from carbonic acid (H_2_CO_3_), shaped through the dissolution of carbon dioxide from the ambient air into water under moist atmospheric circumstances. Otherwise, hydrogen ions in water may originate from the dissolution of other acidic oxides existent in the atmosphere, thus raising cathodic behavior at this site [25].

Galvanic corrosion is displayed when two dissimilar metals launch either physical or electrical contact with one another within a communal electrolytic environment or when a singular metal is opened to electrolytes with fluctuating concentrations. In a galvanic couple, the metal with greater reactivity (the anode) practices accelerated corrosion, while the more inert metal (the cathode) undergoes corrosion at a slower rate. Equally, when immersed independently, each metal corrodes at its own pace. Assortment of suitable metals can be channeled by reference to the galvanic series. For instance, zinc frequently obliges as a sacrificial anode for safeguarding steel structures [26].

### 2.2. Corrosion Prevention Mechanism

The shielding tools engaged by anticorrosion coatings encompass diverse strategies, such as the effects of the type of the coating, reticence, inactiveness, and self-healing. Barrier coatings are aided by impeding the ingress of corrosive agents to the substrate, while coatings comprising zinc or its alloys confer cathodic protection. Furthermore, anodic chemical treatment is achievable for coatings responsive to passivation processes [27]. Another category, intelligent or smart coatings, is designed to selectively release reparative or protective agents to damaged areas upon external stimulus, ensuring active and durable deterioration prevention for the coated metal [28,29,30].

### 2.3. Anticorrosion Coatings

Three main types exist: mineral, inorganic, and organic coatings. Metallic coatings can function as barriers and offer cathodic protection by experiencing galvanic corrosion, depending on their nobility in relation to the base metal [31]. Collective procedures for utilizing the above three mentioned coatings include thermal dropping, silver-plating, hot spurting, shielding, biochemical gas exposure, and superficial amendment using focused energy (laser or ion) rays [5,32].

Inorganic coatings primarily safeguard the implemented substrate via blockade prevention mechanisms; however, using Zn-based coatings enhances cathodic fortification [33]. This class comprises various materials such as hydraulic adhesives, ceramic ware, muds, crystal, carbon, and silicon-based coatings. Chemical alteration coatings, which transform the metal surface into a passive film, improve the corrosion protection enactment, making them active bases for subsequent protective paint applications [32]. The sol-gel derived coatings reveal excellent barrier capability. On the other hand, sol-gel coatings composed of a hybrid mixture of organic and certain inorganic materials offer enhanced performance due to their tractability, corrosion safeguard, and applicable curing condition [34,35,36,37,38,39]. Progressions in the field include the incorporation of graphene, renowned for its waterproofness to corrosive agents and limitation of moisture and ion diffusion, thus stimulating coatings against corrosion [40]. Mixtures of organic–inorganic preparations, predominantly organic–inorganic nanocomposites, have established amplified linkage to substrates, raised thermal stability, and fortified barrier properties, extending the durability of coated materials in perplexing scenarios [41].

The deliberate addition of nanoparticles, bridging metal oxides, clay minerals, and carbon-based materials has suggestively heightened the traits of organic coatings. Nanoparticle-infused coatings unveil perceptible improvements in mechanical strength, barrier physical appearance, and UV resistance, so refining inclusive enactment and protective efficacy across miscellaneous environments [42].

## 3. Methods

### 3.1. Materials

The samples used in this study were prepared from the following materials: Polyaniline (PANI, emeraldine base, 50,000 g/mol), camphor sulfonic acid (CSA, 232.30 g/mol), N-Methyl-2-Pyrrolidone (NMP, 99.133 g/mol), ferric oxide nanoparticles (Fe_3_O_4_NPs, 50–100 nm particle size), cerium dioxide nanoparticles (CeO_2_NPs of size less than 50 nm). All the materials were purchased from Sigma Aldrich (formally Millipore Sigma, St. Louis, MO, USA).

### 3.2. Synthesis Technique

To prepare the PANI composite solution, 0.5 g Polyaniline and 0.12 g CSA were dissolved in 100 mL NMP. The mixture was subjected to vigorous magnetic stirring overnight at 55 °C. To obtain a homogenous solution, the PANI composite solution was sonicated at 55 °C for three hours. Moreover, solution mixed method was utilized to synthesize PANI/CeO_2_ NPs nanocomposite solutions. Under magnetic stirring for 5 h, 5 wt.% of CeO_2_ was added separately to the PANI composite solution at room temperature.

The resulting PANI/CeO_2_ nanocomposite solutions were sonicated for 3 h at room temperature. To obtain the desired investigated structures, we employed the casting technique to deposit PANI and PANI/CeO_2_ nanocomposite films on ITO and steel substrates. The casting process was conducted under ambient conditions, specifically at room temperature and ambient atmospheric pressure. To prevent any modifications on the surface morphology and to ensure complete drying of all residuals, the deposited films were dried at 40 °C under ambient conditions overnight.

### 3.3. Characterization Methods

The chemical, structural, and morphological characterizations were conducted using an FTIR microscope (HYPERION 3000 Bruker), XRD (Malvern Panalytical Ltd., Malvern, UK), and SEM micrographs (Quanta FEG 450), respectively. Thermogravimetric analysis (TGA, NETZSCH) was utilized to investigate the thermal stability of the as-prepared PANI and PANI/CeO_2_ thin films. A UV–Vis spectrophotometer (Hitachi U-3900H) with a total internal sphere was used to obtain and interpret the optical properties. A four-point probe (Microworld Inc., Farmington Hills, MI, USA) hocked up to a high-resolution Keithley 2450 Sourcemeter was employed to measure and interpret the electrical conductivity. The corrosion protection performance was investigated using the polarization method. The contact angle measurement was performed utilizing a custom-built setup comprising a commercial camera focused on the stage within the confocal distance. This experimental arrangement involved capturing images of the pendant drop, with the camera connected to a computer for data acquisition and digitization. A syringe positioned in front of the camera was used to generate a pendant drop suspended from a metallic needle. This description has also been integrated into this manuscript.

## 4. Results and Discussion

The chemical, structural, and morphological characteristics of PANI and PANI/CeO_2_ nanocomposite films were investigated and interpreted using the FTIR absorbance spectra (Figure 1), XRD patterns (Figure 2), as well as SEM and water contact angle measurements (Figure 3). Figure 1 displays the FTIR spectra of PANI and PANI/CeO_2_ nanocomposite films in the 500–4000 cm^−1^ spectral range. For the protonated PANI film, the C=N iminoquinone vibrational band appears at 660 cm^−1^ [23]. Additionally, the absorption band at 825 cm^−1^ is assigned to the aromatic rings. This is strong evidence of the formation of the polymer [24]. The vibrational band that appears at 950 cm^−1^ refers to the $-{SO}_{3}H$ group, ratifying the PANI protonation with CSA. Furthermore, the in-plane C–H bending vibrations within the quinoid unit (N=Q=N) appear at 1125 cm^−1^. Also, the aromatic C–N stretching vibrations appear in the 1300–1500 cm^−1^ spectral range [25]. The C–N stretching vibrations inside benzenoid (N–B–N) and quinoid (N=Q=N) rings are located at 1541 and 1651 cm^−1^, respectively. The absorption bands beyond 3000 cm^−1^ indicate the N–H stretching vibrations [25]. Incorporation of CeO_2_ NPs into the protonated PANI films strongly moved the vibrational bands to a higher region of the spectrum. This strong shift is mainly attributed to the difference in the electronegativity between CeO_2_ NPs and the PANI molecules.

Figure 2 shows the XRD patterns of PANI and PANI/CeO_2_ nanocomposite films in the 10°–50° angular range. The crystal structure of PANI is mainly determined by the synthesis conditions and the type of the protonic acid [26]. The protonated PANI film shows diffraction peaks at 14.97°, 20.72°, and 25.38° corresponding to (011), (001), and (110) diffraction crystallographic planes [27]. Additionally, a diffraction peak located at 27.86° is associated with the CSA, confirming the protonation of PANI. The semi-crystalline protonated PANI with CSA has two phases, namely, the phase in which the polymer chains are ordered (crystalline phase) and the phase in which the polymer chains are randomly distributed (amorphous phase) [28]. On the contrary, the protonated PANI/CeO_2_ nanocomposite film exhibits an amorphous phase.

Figure 3 demonstrates the 1 μm scaled SEM micrographs and water contact angle (WCA) measurements for PANI and PANI/CeO_2_ nanocomposite films. As evident from the observation, the protonated PANI film exhibits a rod-like structure with a water contact angle (WCA) of 38°, suggesting its hydrophilic nature (Figure 3a). This characteristic is known to influence corrosion [29]. Adding CeO_2_ NPs into the PANI matrix leads to the decrease of the grain sizes as well as the WCA (24°) (Figure 3b).

The absorption spectra of PANI and PANI/CeO_2_ nanocomposite films were investigated. The absorption coefficient can be expressed as $\alpha =\left(1/d\right)\ln\left(\left(1-R\right)/T\right)$ [30,31]. The parameters T, R, and d stands for the transmittance, reflectance, and film thickness, respectively. The parameter α exhibits a sudden decrease from 0.01 to 0.003 as the incident wavelength increases from 300 to 350 nm. Beyond λ=350 nm, it attains a constant value as demonstrated by Figure 4. The vibration band that appears between 400 and 480 nm is related to the superposition of the π-π* transition within the benzoin ring with the confined polaron (polaron-π*) transition [32]. Introducing CeO_2_ NPs into the PANI composite matrix increases α in the visible region and shifts the absorption edge into the red region. The bandgap energy of both PANI and PANI/CeO_2_ nanocomposite films was calculated based on the Tauc plot [33,34]. The bandgap energy of the PANI composite film was calculated to be 3.74 eV. Introducing CeO_2_ NPs into the PANI matrix decreases the bandgap energy to 3.67 eV.

Both PANI and PANI/CeO_2_ nanocomposite films were electrically characterized by measuring their electrical conductivity using a four-point probe at 12 distinct points. The measured electrical conductivity of the PANI composite film was determined to be 0.40 S·cm^−1^. This value falls within the range typically reported for HCl protonated polyaniline films, which typically exhibit conductivities ranging from 1.9 to 3.5 S cm^−1^ [35]. Additionally, the data reported in Ref. [36] corroborates this finding. In this study, the PANI-CSA film exhibited a similar electrical conductivity of 0.40 S·cm^−1^. The crystallographic structure of PANI promotes interactions between neighboring polarons, facilitating interchain transitions and enhancing polaron delocalization along the polymer chain. This structural conformation also optimizes the packing of polymer chains into a highly ordered molecular state, thereby favoring higher electrical conductivity compared to localized configurations [25]. This structural conformation also optimizes the packing of polymer chains into a highly ordered molecular state, thereby favoring higher electrical conductivity compared to localized configurations. The high conductivity observed can be attributed to the strong acid-doping of CSA, which introduces additional charge carriers by protonating the imine nitrogen of the PANI backbone [37]. The incorporation of CeO_2_ NPs into the PANI composite film results in a more than twofold increase in electrical conductivity, reaching 1.07 S·cm^−1^. To gain further insight into the electrical conductivity values, electrical conductivity maps (1 cm × 1 cm) of PANI and PANI/CeO_2_ nanocomposite films were generated and are presented in Figure 5. Analysis of Figure 5a shows significant conductivity variation across the PANI composite film, attributed to substantial surface morphology alterations and variations in the quality and conditions of the growth process. The introduction of CeO_2_ NPs into the PANI composite matrix induces notable changes in the conductivity distribution, likely due to the dispersion of nanoparticles within the PANI matrix (Figure 5b).

The polarization method was used to elucidate the corrosion rate of PANI and PANI/CeO_2_ nanocomposite films. The corrosion test was evaluated in 3.5 wt.% NaCl solution at a temperature of 298 K (Figure 6a). The corrosion potential ($E_{corr}$) and corrosion current ($I_{corr}$) were evaluated at the junction where the tangent of the anodic and cathodic polarization curves intersects. The corrosion rate of the nanocomposite films can be calculated using $CR=kMI_{corr}/\rho_{m}$ [38,39], where k is a parameter equal to 3268.5 mol/A, M stands for the molecular weight of carbon steel, and $\rho_{m}$ is the density of carbon steel. The parameters $E_{corr}$, $I_{corr}$, and CR of electrochemical measurements are tabulated in Table 1. The corrosion rate of PANI-coated carbon steel is lower than the corrosion rate of bare carbon. In addition, incorporating PANI with CeO_2_ decreases the corrosion rate. The lowest corrosion rate is found for PANI/CeO_2_ nanocomposite films (0.112 mm/year). The protection efficiency ($\eta_{PROT}$(%)) can be calculated using $\eta_{PROT}\left(\%\right)=\left(i_{corr}^{0}-i_{corr}^{coat}\right)/i_{corr}^{0}\times 100\%$, where the parameter $i_{corr}^{0}$ represents the corrosion current of bare carbon steel and $i_{corr}^{coat}$ stands for the corrosion current of the nanocomposite-coated carbon steel [40]. The higher protection efficiency was obtained for PANI/CeO_2_ nanocomposite films with a value of 92.25%.

The corrosion rates of PANI and PANI/CeO_2_ nanocomposite films were calculated in the 298–338 K temperature range (Figure 6b). As the temperature of the electrochemical reaction is increased, the corrosion rate for all nanocomposite films increases. This increase is a direct consequence of the enhanced electrochemical reaction rates as well as the increase in the kinetic energy of the molecules in the electrolyte solution. Therefore, the diffusion rate of the molecules is significantly increased. To clarify the thermal activated processes of the corrosion reactions [41], the corrosion rates of PANI and PANI/CeO_2_ nanocomposite films versus the reciprocal of temperature (1000/T[K]) are illustrated in Figure 6c. As can be seen, corrosion rates exhibit Arrhenius-like behavior ($CR=CR_{0}\exp\left(-E_{a}/K_{B}T\right)$) [42]. This means that the corrosion rate of bare carbon steel and carbon steel coating by PANI, PANI/ZrO_2_, PANI/Fe_3_O_4_, and PANI/(ZrO_2_-Fe_3_O_4_) nanocomposite films are thermally activated. The activation energies deduced by Arrhenius fitting are tabulated in Table 1. Obviously, the highest activation energy is obtained for the PANI/CeO_2_ nanocomposite film. Thus, the effect of temperature on the corrosion rate of the PANI/CeO_2_ nanocomposite film is more pronounced than for other investigated samples.

Figure 7 illustrates the effect of coating on the Nyquist and the Bode plots of PANI and PANI/CeO_2_ nanocomposite films immersed in 3.5 wt.% NaCl at 298 K. In the Nyquist plot, the curves have single capacity arcs with large radii for PANI/CeO_2_ nanocomposite films compared to bare carbon steel and PANI film (Figure 7a), indicating the resistance of these nanocomposites as very high. The coating has an exceptional physical barrier influence on the electrolyte. The impedance value (Z) depicted in the Bode diagram at a frequency of 0.1 Hz demonstrates an increase from 652 Ω·cm^−2^ for the bare steel to 1900 Ω·cm^−2^ for the PANI/CeO_2_ nanocomposite film. Additionally, for the PANI/CeO_2_ film, the impedance value is recorded at 1930 Ω·cm^−2^. Thus, coating the carbon steel with the PANI/CeO_2_ nanocomposite film enhances the corrosion resistance (Figure 7b). 

## 5. Conclusions

This work reports the results of a novel nanocomposite coating of PANI/CeO_2_ nanocomposite film for corrosion protection applications. FTIR spectra and XRD patterns confirm the formation of the nanocomposite films. Water contact angle measurements confirm that the PANI/CeO_2_ nanocomposite film has a hydrophilic nature. Therefore, the films can interact with water through hydrogen bonding. Optical measurements reveal that the bandgap energy of the PANI composite film is 3.74 eV. Introducing CeO_2_ NPs into the PANI matrix decreases the bandgap energy to 3.67 eV. Consequently, bandgap engineering in the PANI/CeO_2_ nanocomposite film could be a powerful technique for the design of new materials and devices based on this novel material. In addition, band diagrams with continuous bandgap variations can be generated in heterojunctions designed from this novel material and fabricated using techniques such as molecular beam epitaxy. The electrical conductivity measurements reveal that the PANI composite film exhibits an electrical conductivity of 0.40 S·cm^−1^. Introducing CeO_2_ NPs into the PANI composite film increases the electrical conductivity by more than twofold to 1.07 S·cm^−1^. The corrosion protection performance was investigated using electrochemical measurements. The protection efficiency of the PANI/CeO_2_ nanocomposite film is 92.25%. In general, the efficiency of an inhibitor increases with an increase in the inhibitor concentration. Obtaining excellent inhabitation for incorporating 5 wt.% of CeO_2_ in PANI nanocomposite films is very promising for corrosion protection applications.

## Figures and Tables

**Figure 1 nanomaterials-14-00526-f001:**
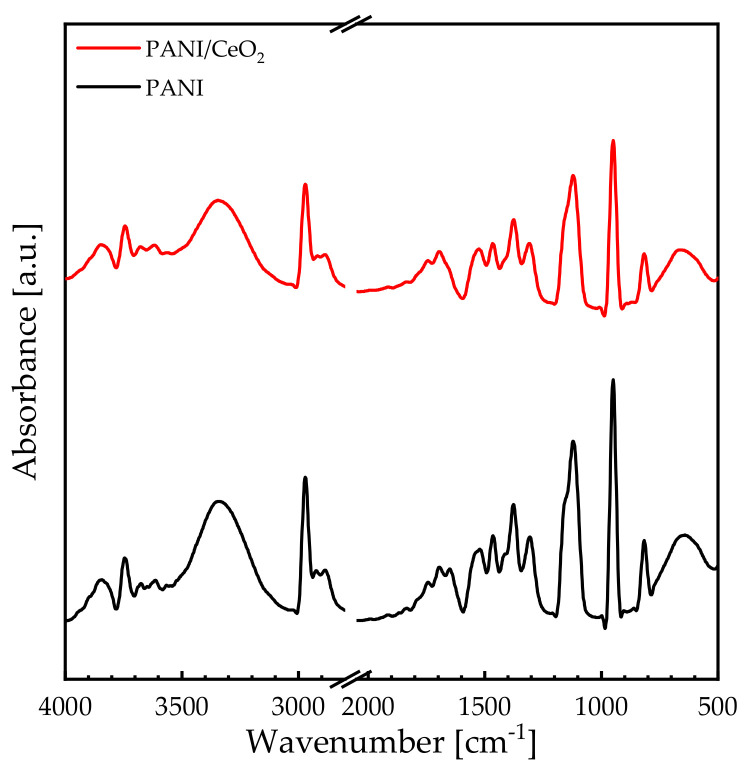
The FTIR spectra of PANI and PANI/CeO_2_ nanocomposite films in the 500–4000 cm^−1^ spectral range.

**Figure 2 nanomaterials-14-00526-f002:**
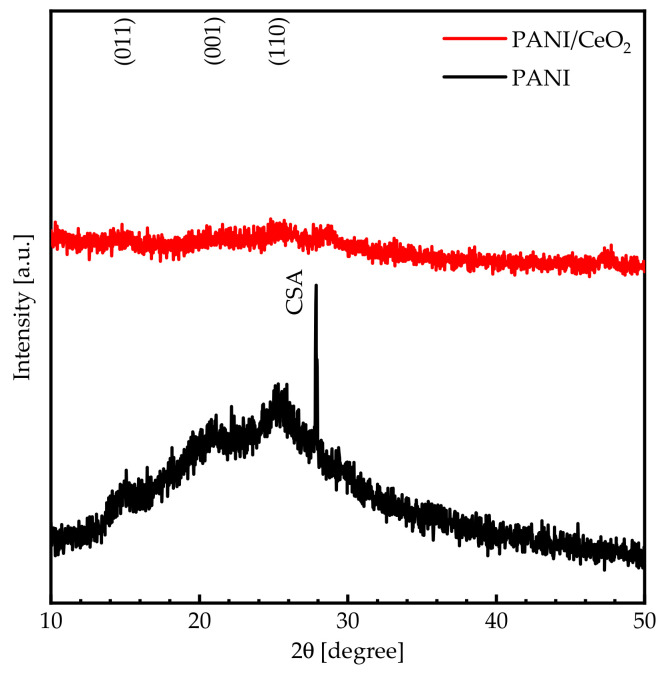
The XRD patterns of PANI and PANI/CeO_2_ nanocomposite films in a diffraction angle range of 10°–50°.

**Figure 3 nanomaterials-14-00526-f003:**
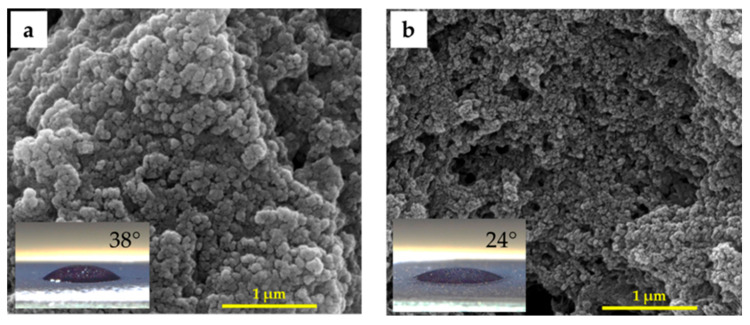
SEM images of (**a**) PANI and (**b**) PANI/CeO_2_ nanocomposite films at 1 μm scale.

**Figure 4 nanomaterials-14-00526-f004:**
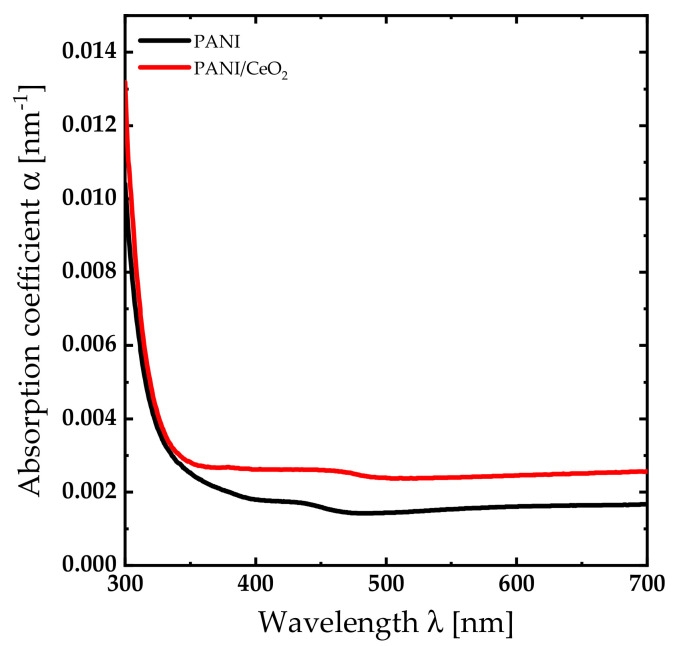
Absorption coefficient spectra of PANI and PANI/CeO_2_ nanocomposite films.

**Figure 5 nanomaterials-14-00526-f005:**
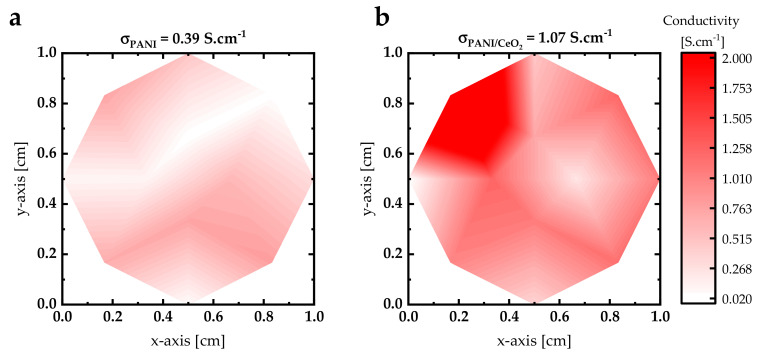
The electrical conductivity maps of (**a**) PANI and (**b**) PANI/CeO_2_ nanocomposite films.

**Figure 6 nanomaterials-14-00526-f006:**
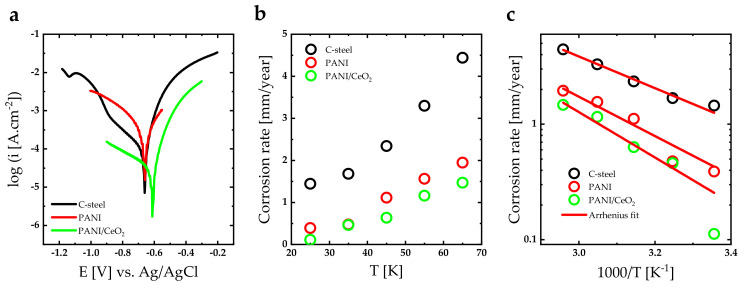
(**a**) Tafel curve of carbon steel coated with PANI and PANI/CeO_2_ nanocomposite films at 298 K, (**b**) corrosion rate of carbon steel coated with PANI and PANI/CeO_2_ nanocomposite films as a function of temperature [K], and (**c**) corrosion rate of carbon steel coated with PANI and PANI/CeO_2_ nanocomposite films as a function of 1000/temperature [K] fitted to Arrhenius function.

**Figure 7 nanomaterials-14-00526-f007:**
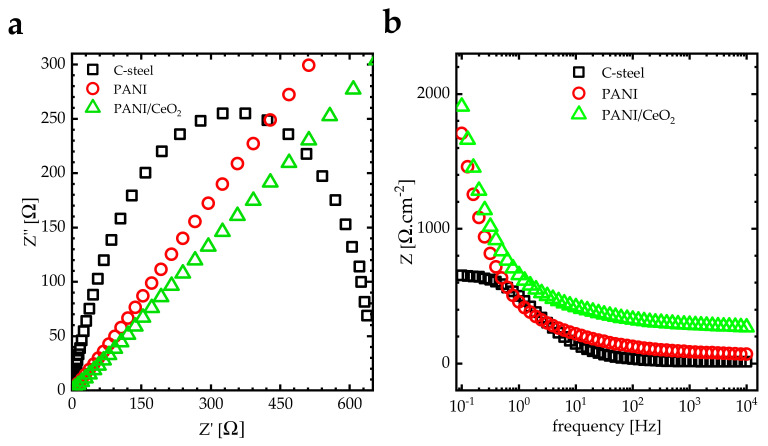
(**a**) Nyquist and (**b**) Bode plots of carbon steel coated with PANI and PANI/CeO_2_ nanocomposite films at 298 K.

**Table 1 nanomaterials-14-00526-t001:** Electrochemical parameters values for carbon steel coated with PANI and PANI/CeO_2_ nanocomposite films at 298 K values calculated from Tafel plots.

	C-Steel	PANI	PANI/CeO_2_
CR [mm/year]	1.445	0.390	0.112
$\eta_{PROT}$ (%)	-	72.90	92.25
$E_{a}$ [eV]	0.27	0.34	0.39

## Data Availability

The data that support the findings of this study are available on request from the corresponding author [Ahmad Alsaad]. Unprocessed data are available upon request from the corresponding author.

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
