# Peer review of "Synthesized PANI/CeO2 Nanocomposite Films for Enhanced Anti-Corrosion Performance"

_nanomaterials, 2024, doi:10.3390/nano14060526_

Round 1

Reviewer 1 Report

Comments and Suggestions for Authors

Nanomaterials

Characterization of As-synthesized PANI/CeO2 Nanocomposite Films for Enhanced Anti-Corrosion Applications.

The authors synthesized PANI/CeO2 nanocomposite films, analyzed through FTIR spectra and XRD patterns confirming the successful formation of the nanocomposite films. Also, they evaluated its corrosion protection efficiency.

I suggest to the authors:

1.    Review punctuation.

2 . Methods

2.1 . Materials

2.    5 wt.% of CeO2  instead five wt.% of CeO2. 

3.    Indicate experimental conditions of casting technique.

4.    It is not clear when was used ITO or steel substrate.  

5.    It was not indicated the procedure to contact angle measurements. 

6.    Indicate how were the parameters 𝑇𝑅, and 𝑑 determined.

7.    Detail the Tauc plot method. What was the n value used and why? 

8.    Expand discussion of the electrical conductivity maps, how were they obtained, etc. 

9.    The parameters 𝐸corr and 𝐼corr are not tabulated in Table 1.  

10.         Include Arrhenius function settings. 

11.      In the Nyquist plot, it is not evident that the capacity arc with large radii correspond to PANI/CeO2 nanocomposite.

12.                 The Z value in the Bode diagram at 0.1 Hz increases from 652 Ω.cm-2 for bare steel to 1900 Ω.cm-2 for PANI/CeO2 nanocomposite film. ALSO, CAMPARE WITH PANI. 

13.                 Make a comparison with the results reported with other PANI/oxides. 

14.                 Determine de effect of time in anti - corrosion behavior.

Comments on the Quality of English Language

Minor editing

Author Response

Dear Reviewer 1,

Attached is the point-point rebuttal which includes the response to each point and suggestion provided by Reviewer 1.

Please accept my best regards,

Prof. Dr. Ahmad Alsaad

Reviewer 2 Report

Comments and Suggestions for Authors

This article deals with formation a novel nanocomposite coating of PANI/CeO2 filma as corrosion protection compounds.  The authors should reduced word - sentence "clear" , "clearly seen" etc..

Fig.3  Both surfaces on images a, b, are hydrophillic; SEM on fig. b is little more as PANI alone; those the corrosion resistance  is nearly the same or very similar; it requires  next interpretation.

The sentnece in RandD p2., "SEM images and water contact angle  measurements (Fig.3) " seems ownerless; it needs more details; the same situation is with FTIR results and others. 

I can recommend this article for publication after major, very  major revision.

Author Response

Dear Reviewer 2,

Attached is the point-point rebuttal which includes the response to each point and suggestion provided by Reviewer 2.

Please accept my best regards,

Prof. Dr. Ahmad Alsaad

Reviewer 3 Report

Comments and Suggestions for Authors

The authors researched the anti-corrosion performance of PANI/CeO2 composite coating in detail. Besides, the authors innovatively studied the electrical conductivity of the coatings, it is quite interesting. It is a good work with a minor revision.

1.       In the introduction part, more nano particles should be added and discussed.

2.       The results in Fig.7a, the plots look like not whole displayed. Please modified it.

3.       Some revelant works should be cited in relevant position, such as: Prog. Org. Coat. 2024, 188, 108250.

4.       The language and grammar of this manuscript should be checked carefully.

Comments on the Quality of English Language

Should be improved

Author Response

Dear Reviewer 3,

Attached is the point-point rebuttal which includes the response to each point and suggestion provided by Reviewer 3.

Please accept my best regards,

Prof. Dr. Ahmad Alsaad

Round 2

Reviewer 1 Report

Comments and Suggestions for Authors

I have reviewed the new version of the article and I agree with the corrections made, however I consider that section 2 should not be included since its contents are well documented in the literature.

Author Response

Dear Reviewer 1,

Reviewer comment:  have reviewed the new version of the article and I agree with the corrections made, however I consider that section 2 should not be included since its contents are well documented in the literature.

Authors Reply: Thank you for approving the corrections we made in the first round of revision. Regarding section 2, the editorial office asked us to increase the number of words in the manuscript. The number of words should be above 4000 words to consider the submission of the manuscript as an article. If the number of words is much less than 4000 words, the submission would be consider as a communication. Section 2 discusses corrosion protection and since PANI/CeO2 Nanocomposite Films investigated in this manuscript are found to be potential candidates for Enhanced Anti-Corrosion Performance, we would highly appreciate if you could approve this section in the manuscript. Again thank you for enriching the contents of this manuscript.

Reviewer 2 Report

Comments and Suggestions for Authors

No comments to revised version.

Author Response

Dear Reviewer,

Thank you very much for approving the revision of the first round.

Best regards,